# A Lighted Deep Convolutional Neural Network Based Fault Diagnosis of Rotating Machinery

**DOI:** 10.3390/s19102381

**Published:** 2019-05-24

**Authors:** Shangjun Ma, Wei Cai, Wenkai Liu, Zhaowei Shang, Geng Liu

**Affiliations:** 1Shaanxi Engineering Laboratory for Transmissions and Controls, Northwestern Polytechnical University, Xi’an 710072, China; nwpucaiwei@mail.nwpu.edu.cn (W.C.); npuliug@nwpu.edu.cn (G.L.); 2Key Laboratory of Dependable Service Computing in Cyber Physical Society Chongqing University, Chongqing 400044, China; kev@cqu.edu.cn (W.L.); szw@cqu.edu.cn (Z.S.)

**Keywords:** deep learning, wavelet packet transform, convolutional neural networks, fault diagnosis, rotating machinery

## Abstract

To improve the fault diagnosis performance for rotating machinery, an efficient, noise-resistant end-to-end deep learning (DL) algorithm is proposed based on the advantages of the wavelet packet transform in vibration signal processing (the capability to extract multiscale information and more spectral distribution features) and deep convolutional neural networks (good classification performance, data-driven design and high transfer-learning ability). First, a vibration signal is subjected to pyramid wavelet packet decomposition, and each sub-band coefficient is used as the input for each channel of a deep convolutional network (DCN). Then, based on the lightweight modeling requirements and techniques, a new DCN structure is designed for the fault diagnosis. The proposed algorithm is compared with the support vector machine algorithm and the published DL algorithms based on a bearing dataset produced by Case Western Reserve University. The experimental results show that the proposed algorithm is superior to the existing algorithms in terms of accuracy, memory space, computational complexity, noise resistance, and transfer performance, producing good results.

## 1. Introduction

Rotating machinery systems have been extensively applied in a number of engineering fields (e.g., aviation, ships and warships, machine tools and vehicles) and play an increasingly pivotal role. Rotating machinery damage and faults not only severely affect the reliability and safety of the entire system but also cause tremendous economic losses. Therefore, researchers have been continuously conducting relevant research. One current focus of research is to extract and classify fault features, i.e., to further improve the identification accuracy and system monitorability by analyzing initial weak fault signals to achieve real-time monitoring and diagnosis.

Based on the differences in the feature extraction and fault diagnosis algorithms, rotating machinery fault diagnosis algorithms can be classified into two types, namely, vibration analysis and intelligent diagnosis [1,2,3,4,5]. In a vibration analysis, signal decomposition techniques (e.g., wavelet transform (WT) [6] and empirical mode decomposition (EMD)) are used to directly detect fault frequencies in the original data [7]. Most fault frequencies are hidden in low-frequency deterministic components and high-frequency noise components and are consequently very difficult to observe in the spectra. As a result, vibration analysis algorithms have a relatively poor practical performance. Intelligent diagnosis is a new research direction and includes artificial neural networks (ANNs) and support vector machines (SVMs) as the main algorithms. Rojas et al. [8] extracted 32-dimensional features from signals using the Fourier transform and classified them using an SVM. Xia et al. [9] proposed to use the Volterra series as a feature for describing the operating conditions of machinery systems and used a backpropagation neural network for classification and discrimination. Lei et al. [10] proposed to extract features using EMD combined with wavelet packet decomposition (WPD) and input selected sensitive features into an ANN for a fault diagnosis. Gangsar et al. [11] proposed to extract fault features from signals using the WT and classify them using an SVM. While conventional intelligent diagnosis algorithms have been extensively applied in machinery signal fault diagnosis, they have the following disadvantages [12,13,14]. (1) The accuracy of the fault diagnosis is dependent on the quality of the feature extraction. Vibration signals collected in an industrial environment are always complex, unstable and contain high levels of noise. Thus, to ensure the quality of the feature extraction, a manual design and suitable features are needed based on the characteristics of the different faults. The quality of the features directly determines the system performance. Therefore, the system feature extraction is not automatic. (2) Conventional intelligent diagnosis algorithms are based on shallow learning models, which are unable to effectively learn nonlinear relations in complex systems.

To address these deficiencies, deep learning (DL) has been gradually used in machinery signal fault diagnosis. Based on the model used in the DL, the available studies are classified into three types, namely, those involving autoencoders (AEs), convolutional neural networks (CNNs) and recurrent neural networks (RNNs) [4]. Jia et al. [15] pretrained a three-layer deep neural network (DNN) by stacking AEs and obtained final prediction results by fine-tuning the DNN. Li et al. [16] proposed a deep random forest fusion structure, which extracts signal features using the WT in conjunction with a deep Boltzmann machine and produces classification results by feature fusion using a random forest. Gan et al. [17] extracted the wavelet packet energy of the original signal as the feature input and then established a layered deep belief network-based diagnostic network, with its first layer diagnosing the fault type and the second layer determining the severity of the fault. Relying on network optimization using sparse regularization, Sun et al. [18] proposed an algorithm for predicting motor faults. This type of algorithm can be relatively easily realized and can obtain feature representation by learning, but it is slow in convergence and has a weak transfer-learning ability. The main RNN algorithms include RNNs and long short-term memory neural networks (LSTMNNs). Zhao et al. [19] proposed an LSTMNN-based fault diagnosis algorithm. This type of algorithm is effective in detecting faults in time-series data and is able to discover problems that arise as time elapses, but it has a relatively high network complexity and a weak transfer-learning ability.

There are a gradually increasing number of studies involving deep CNNs (DCNNs) in the fault diagnosis field. Abdeljaber et al. [20] established a one-dimensional (1D) LeNet5 network based on the LeNet5 architecture for detecting structural damage in machinery. DCNN models that differ in structure have also been proposed, based on 1D CNNs, for predicting faults in various types of rotating machinery [21,22,23,24]. However, most of these models use the original signal as the input but neglect its frequency-domain characteristics. DCNN models that differ in structure have been proposed based on CNNs for fault diagnosis [25,26,27]. In [25], a 1D problem was converted into a two-dimensional (2D) signal processing problem using a continuous WT scalogram. In [26,27], a 1D problem was converted into a 2D signal-processing problem by a non-overlapping equal division of the original signal, before a fault prediction was performed using a 2D convolution algorithm. Zhao et al. [28] proposed to use a wavelet packet-residual network hybrid algorithm for predicting faults in gearboxes. This type of algorithm is effective in analyzing multidimensional data and can effectively extract local features. Compared to 1D processing, 2D representation has a relatively complex structure, and 2D processing requires more time and computational resources in the training and testing processes. For example, the computational load of a 3 × 1 convolution is less than one-third that of a 3 × 3 convolution.

While the available DL algorithms are effective, the collection of original signals in an actual industrial production environment is relatively significantly affected by various components, requiring that the algorithms be highly resistant to disturbances. In addition, with the development of the industrial internet of things (IIoT), the installation of a health management unit in rotating machinery has become a trend. Constrained by costs, as well as by actual hardware and environmental conditions, the improvement of the functional intelligence level of software has high requirements for operating environment resources. As a result, the requirements for the computational complexity of DL algorithms and the space occupied by models are becoming increasingly stringent. Consequently, there are increasingly higher “small-size, lightweight, and rapidness” requirements for DL algorithms [25,29]. The complex and varied conditions of rotating machinery (with significant changes in the rotational speed and load, particularly when the system starts and shuts down) result in unstable collected data. As the conditions change and time elapses, the sample distribution no longer meets the same distribution requirement, leading to a demand for a high transfer-learning ability. The available studies are deficient in lightweight DL (as shown in Table 1), cannot meet the new requirements of the IIoT, and fail to sufficiently consider the noise resistance and transfer-learning ability of algorithms.

Hence, in this study, a lightweight network structure is proposed based on the advantages of the WT and DL. The main contributions of this study are as follows.

(1) A 1D CNN structure is proposed that can effectively improve the identification accuracy and that has a relatively high transfer-learning ability and noise resistance.

(2) The proposed network structure is relatively lightweight and has a high computational speed, and it occupies little space.

The paper is organized as follows: Section 2 elaborates the theoretical basis of the WPT and DCN for a fault diagnosis based on the proposed method; The lightweight CNN structure design is described in Section 3; Experiments and analysis are implemented based on a bearing dataset produced by Case Western Reserve University in Section 4 to verify that the proposed algorithm is superior to the existing algorithms; Finally, the concluding remarks are given in Section 5.

## 2. Theoretical Basis

### 2.1. Wavelet Packet Transform (WPT)

The framework of a wavelet packet transform is the extension of the wavelet transform [30]. The wavelet packet function is also a time-frequency function and can be described as:(1)$$w_{i,j}^{n}=\sqrt{2}w^{n}(2^{j}t-k)$$
where *j* and *k* are integers; they are the index scale and translation operations, respectively. The index *n* is an operation modulation parameter or oscillation parameter. The first two wavelet packet functions are the scaling and mother wavelet functions:(2)$$w_{0,0}^{0}(t)=\varphi (t)$$
(3)$$w_{0,0}^{1}(t)=\psi (t)$$

When *n* = 2, 3, …, the function can be defined by the following recursive relationships:(4)$$w_{0,0}^{2n}(t)=\sqrt{2}\sum_{k}h(k)w_{1,k}^{n}\left(2t-k\right)$$
(5)$$w_{0,0}^{2n+1}(t)=\sqrt{2}\sum_{k}g(k)w_{1,k}^{n}\left(2t-k\right)$$
where h(k) and g(k) are the quadrature mirror filters (QMFs) associated with the predefined scaling function and the mother wavelet function, respectively. The wavelet packet coefficients, $w_{i,j}^{n}$ are computed by the inner product <f(t)
$w_{i,j}^{n}$>, which can be defined as:(6)$$w_{i,j}^{n}=\int f(t)w_{i,j}^{n}(t)dt$$

The framework of the WPT algorithm broken up into three resolution levels is shown in Figure 1, and the structure of WPT is a Perfect Binary Tree.

WPT can further obtain the detail wavelet coefficients of the signal in the high frequency area and provides a more detailed and comprehensive time-frequency plane tiling than the discrete wavelet transform (DWT) does [31]. As shown in Figure 2, the many advantages of WPT are used in the discrete signal processing, such as the fault diagnosis of rotating machinery [11] image processing [32] and video processing [33].

### 2.2. Deep Convolutional Networks (DCNs)

DCNs, proposed by LeCun, are an important branch of DL [34,35]. DCNs consist of convolutional and pooling layers as well as activation and loss functions. The convolutional layers use various types of convolution kernels and perform convolution operations on the input signals. Convolution operations are translation invariant for 1D signals and can support neurons in learning features with a relatively high robustness [36].

The pooling layers perform down-sampling operations, which use a specific value as the output value in each specific small area. Generally, the maximum or mean value is used as the output value. Down-sampling operations are performed to non-linearize 1D signals.

The activation functions stimulate neurons based on a series of input values, weights of interneuron connections and activation rules. The loss functions in a CNN are used to evaluate the difference between the output and the actual value at the training stage. Afterwards, the values of the loss functions are used to update the weights between the neurons. The purpose of the training is to minimize the values of the loss functions.

A batch normalization (BN) layer normalizes the data input into each layer of the network (normalized to a mean of 0 and a variance of 1). The BN layers not only improve the gradient flow in the network, allow a higher learning rate, and improve the training speed, but they also reduce the strong dependence on initialization.

### 2.3. One-by-One Convolutions

A 1 × 1 convolution filter is the same as a normal filter except that it has a size of 1 × 1 and does not consider inter-information relationships in the local areas of the previous layer. One-by-one convolutions first appeared in the Network in Network paper and were used to deepen and widen network structures by reducing the dimensionality in inception networks [37]. To reduce the computational complexity of a system, a 1 × 1 convolution can be used to reduce the dimensionality to compress the channels. In [38], the effect of a 1 × 1 convolution with a different proportion of channel numbers on the performance is discussed and verified, and a squeeze ratio of 0.5 was found to be relatively satisfactory after careful consideration. Here, a two-layer convolutional network is presented in Figure 3 as an example. Figure 3a shows a dimensionality reduction without performing a 1 × 1 convolution, and Figure 3b shows a dimensionality reduction by performing a 1 × 1 convolution.

The effect of a 1 × 1 convolution with different proportions of channel numbers on the performance is discussed and verified.

## 3. Lightweight CNN Structure Design

For 1D fault signal classification problems, an end-to-end network structure (as shown in Figure 4) is proposed. Based on the function, this network can be divided into two layers. The first layer performs the wavelet packet transform (WPT), with an aim to extract finer information from a frequency-domain perspective, and the second layer is the designed relatively lightweight CNN.

### 3.1. DCNN Design

The computational resources in a DCNN are affected by the number of channels, kernel size, data length and connection mode. To improve the ability to predict faults in the rotating machinery, different numbers of channels and structures are used in the CNN partly based on the main factors affecting the network convolutions. In this study, seven DCNs are designed for the 1D convolutional networks and their computational loads and storage space (as shown in Table 2). The number of channels and the kernel size differ among the different structures (models). In Table 2, K/C represents the kernel size/number of channels (e.g., 11 × 1/4 represents a kernel size of 11 × 1 and 4 kernels (channels)). In network design, deepening the network can enhance its ability to learn features and can reduce the number of network parameters and the training load.

As shown in Figure 5, the computational complexity and number of parameters of network structure model 2 are 16% and 9% lower, respectively, than those of model 1.

The WPT-CNN structure is described as follows.
The conv_1 to conv_5 convolutional layers of the network include convolution operations, rectified linear unit (ReLU) activation functions and BN (as shown in Figure 6). The use of the ReLU activation functions and BN can increase the rate of convergence and prevent a gradient explosion and vanishing problems. Maximum pooling is used in the poo1_1 to pool_4 pooling layers to allow the features that pass through the pooling layers to be translation invariant, thereby removing some noise while reducing the dimensionality of the features.The lengths of the conv_1 to conv_5 convolutional layers of the network are set to 11, 3, 3, 3 and 3, respectively, using the smoothing function of the convolution kernel. The length of the conv_1 layer is set to 11 to suppress noise.An alternating cascade of convolutional layers with kernel sizes of 3 × 3 and 1 × 1 is used in the conv_3 to conv_5 convolutional blocks to reduce the dimensionality while increasing the depth of the network, thereby reducing the number of training parameters and realizing a lightweight model. This method has been proven effective elsewhere [29].In the second half of the network, the conv_6 and pool_6 layers are used to substitute for a commonly used fully connected layer to reduce the risk of overfitting caused by a fully connected layer while reducing the number of network parameters. A linear activation function is used in the conv_6 convolutional layer because a linear mapping is required to map the number of channels of the input feature to the same value of the classification. Other activation functions are not able to achieve this and may even slow down the convergence of the network. Global average pooling is used in the pool_5 pooling layer to allow the feature map input into each channel to correspond to each output feature, thereby improving the consistency between the feature maps and output types. The stability of the pooling process is also enhanced by the summation of the spatial information.

Here, the example shown in the network map is used for the analysis. The signal has 16 dimensions and 16 channels after passing through the pool_5 pooling layer. After the feature translation, 16 × 16 = 256 dimensions are obtained. A fully connected layer is then needed to reduce the dimensionality to 10 for the classification. Thus, a total of 256 × 10 = 2560 parameters need to be trained. However, after the conv_6 and pool_5 layers are used to substitute for the fully connected layer, only the parameters in the conv_6 layer (a total of 1 × 16 × 10 = 160 parameters) need to be trained, whereas no parameters in the pool_6 layer need to be trained. Hence, the number of parameters decreases 16-fold from 2560 to 160, thereby saving the memory space occupied by the model parameters.

5. The cross-entropy loss is used as the loss function, as shown below:(7)$$Loss(p,q)=-\sum_{x}p(x)\log q(x)$$
where p(x) is the label of the training set and q(x) is the label value predicted by the network. 

In classification problems, the cross-entropy function is often used as a loss function because the gradient of the cross-entropy loss is only related to the correct classification prediction results in the model optimization process. In this way, updating the network parameters only increases the correct classification but does not affect other classifications.

6. Random number initialization

In DL, each convolution kernel is initialized with a random value, which has a certain impact on the training and system performance of the network. In this study, a Xavier initialization is used to promote the convergence of the model [39]. The weight $w=R^{d_{in}\times d_{out}}$ is initialized using the following equation:(8)$$w∼u[-\frac{\sqrt{6}}{\sqrt{d_{in}+d_{out}}},+\frac{\sqrt{6}}{\sqrt{d_{in}+d_{out}}}]$$
where *u* [*a*, *b*] is a uniform sample of the range [*a*, *b*].

### 3.2. Performance Analysis of Network Parameters

The proposed WPT-CNN hybrid network structure is a lightweight network structure that has significant advantages over some commonly used network structures in terms of the computational load and number of parameters. In other network structures, convolutional layers and fully connected layers are the parts that participate in the computation of the floating numbers and contain parameters that need to be trained. The calculation equations are as follows:(9)$$Parameters_{conv}=K_{h}\times K_{w}\times C_{in}\times C_{out}+C_{out}$$
(10)$$Params_{sfc}=I\times O+O$$
(11)$$FLOP_{conv}=2\times H\times W\times K_{h}\times K_{w}\times C_{in}\times C_{out}$$
(12)$$FLOP_{sfc}=2\times I\times O$$
where *Parameters_conv_* and *FLOP_conv_* represent the value of the number of parameters and the computation of the floating numbers in the convolutional layer; *Params_sfc_* and *FLOP_sfc_* represent the values of the number of parameters and the computation of floating number in the fully connected layer, respectively; H, W and *C_in_* are the height, width and number of channels of the input feature map, respectively; *K_h_* and *K_w_* represent the size of the convolution kernel; *C_out_* is the number of convolution kernels; *I* is the dimensionality of the input; and O is the dimensionality of the output.

Various network structures, including WPT-CNN, DNN, 1D LeNet-5, residual network ResNet-18, ResNet-50 and Visual Geometry Group (VGG)-16, are compared in terms of the number of parameters and the computation of the floating numbers. DNN is the deep neural network proposed in [15] for a machinery fault diagnosis, which is pretrained by stacking AEs. DNN contains three hidden layers, each of which contains 600,200,100 neurons. 1D LeNet5 is a 1D CNN that is obtained by improving the LeNet-5 and that is used for motor fault detection. Resnet-18, Resnet-50 and VGG-16 are commonly used CNN frameworks. To accommodate 1D input data, 2D convolutions in these three networks were altered to 1D convolutions for calculation. The input length was determined by the input length of the methods reported in the literature and is an evaluation basis for the computation complexity and number of parameters (see Table 3).

The calculation results show that the proposed algorithm has the lowest computation of floating number computational load (3–500 times lower than that of the available algorithms) and the fewest number of parameters (3–2000 times fewer than those of the available algorithms). In addition, the proposed hybrid network contains 14 convolutional layers and is thus much deeper, and it has a higher ability to extract features than some of the available algorithms (e.g., [23,24,25,26,27]). See the subsequent comparison experiments for details.

## 4. Experiment and Analysis

The hardware and software environments used to implement the algorithm are shown in Table 4.

### 4.1. Experimental Data

#### 4.1.1. Introduction of the Dataset

The bearing dataset collected by the Case Western Reserve University (CWRU)’s Bearing Data Center were used to validate the effectiveness of the proposed WPT-CNN network structure. Figure 7 shows the experimental platform for the data collection [40]. The dataset contains the data for bearings under four different conditions, that is, normal bearings, bearings with a faulty ball (ball), bearings with a faulty inner race (inner_race) and bearings with a faulty outer race (outer_race). For each type of fault, there are three fault diameters, that is, 0.007 in., 0.014 in. and 0.021 in. Thus, with the inclusion of normal bearings, this dataset contains the data for 10 classification groups of bearings (as shown in Table 5).

#### 4.1.2. Data Preparation

Because their volume was limited, the experimental data were augmented using the overlap sampling method reported in [15,24], as shown in Figure 8. In this study, the overlap length was determined based on the data length.

In the experiments, the dataset was divided into eight training sets, one validation set, and one test set. In addition, 800 training samples, 100 test samples and 100 validation samples were randomly selected. In the training process, the validation set was used to examine the identification accuracy after every 10 epochs. 

As shown in Table 5, the dataset was divided into four sub-datasets based on the number of loads. These are A, B, C and D, corresponding to the data collected under 0, 1, 2 and 3 loads, respectively. The datasets A, B, C and D each contain 800 training samples and 100 test samples of each type, totaling 8000 training samples and 1000 test samples.

#### 4.1.3. Experimental Description

The CWRU dataset was collected in an environment with a relatively low level of ambient noise, and there are a number of noise sources in an actual environment. Therefore, in order to reflect the ability of the algorithm proposed in this paper, noise was added into the samples in the original test set. 

A 10%, 30%, 50%, 70%, 90% and 100% white Gaussian noise signal was added to the data to simulate actual conditions. The signal-to-noise ratio (SNR) was used when adding the noise signal and is defined as follows:(13)$$\mathrm{SNR}=10\log_{10}P_{signal}/P_{niose}$$
where *P_signal_* and *P_niose_* are the intensity of the original and noise signals, respectively. 

Figure 9 shows the original vibration signal from a bearing with a 0.007 in. faulty inner race under 0 load and the signals with various percentages of added noise. Note that the data with 0% noise represent the results under normal conditions.

The Daubechies-1 wavelet function was selected for all experiments. The filter of this function has various lengths and a low smoothing effect, which is favorable for the extraction of detailed signal features.

Various types of noise are present in an industrial production environment. Therefore, fault prediction algorithms must be relatively highly resistant to noise. Because the number of loads and the operating rotational speed of a machine change when it is in operation, the distribution of the collected sample data varies between different conditions. Therefore, algorithms are required to have a relatively high transfer-learning ability for various loads.

To examine the noise resistance of the proposed algorithm, the learning model with the highest accuracy for the validation set in 1000 iterations was selected. Noise-containing data were randomly generated 10 times for each sample in the training set of dataset A and used for testing and statistically analyzing the experimental results, which were represented in the form of the mean and standard deviation. Similarly, dataset A was used as a training set to train the model, and datasets B, C and D were used to test the accuracy of the algorithm (denoted by AB, AC and AD, respectively) in order to determine the transfer-learning ability for various loads.

### 4.2. Experiment and Analysis

**Experiment 1:** The effect of the different number of decompositions of WPT on the accuracy and noise resistance of the algorithm.

In this section, to determine the best number of decompositions of WPT, the network structure model 2 with three convolution kernels in the first convolutional layer is selected. The accuracy of the proposed network structure in the time domain and the different number of decompositions of WPT under different percentages of noise are obtained for the validation and test sets. Table 6 summarizes the statistical experimental results.

Because the original signal was used as the input signal, the input signal had a length of 1024. In WPT, as the decomposition level increases, the length of each sub-band decreases by a factor of 2, whereas the number of sub-bands increases by a factor of 2. For example, when decomposed at levels 3, 4 and 5, the length of each sub-band is 128, 64 and 32, and there are 8, 16 and 32 sub-bands, respectively. The experimental results show that for a 1D input, a sub-band length of 64 resulted in a satisfactory division of the frequency domain, a moderate length and a large amount of information.

**Experiment 2:** Effects of various sizes of convolution kernels in the first layer on the noise resistance of the algorithm.

Table 6 shows that the number of decompositions of the WPT level is 4, which has the highest noise resistance, so the WPT 4 and network structure model 2 are thus selected in this section. An experiment was conducted to compare the effects of the various sizes of convolution kernels in the first layer. Table 7 summarizes the experimental results.

The experimental data in Table 7 show that when taking the computational load and performance into account, a first-layer convolution kernel size of 11 produced a satisfactory noise-suppressing effect.

**Experiment 3:** Comparison of the performance of three different CNN models

The effects of the number of channels on the noise resistance and transfer-learning ability of the network structures were comparatively analyzed. The data with 0% noise represents the results under normal conditions.

As demonstrated in Table 8, model 2 was superior to model 1, with the same number of channels in each metric, suggesting that a 1 × 1 convolution can help improve the performance.

The number of channels of each model can be set to 4, 8, 16, 32, 64, and 128. When there are 4 or 8 channels, to ensure that there are sufficient channels for depicting features, a dimensionality reduction is generally not performed. When there are a large number of channels (e.g., 16, 32, 64 and 128), a dimensionality reduction can be performed but does not significantly improve the performance. This is because inter-elemental relationships in a 1D signal are simple compared to a 2D signal, and because an excessive number of channels cannot facilitate an improved performance. However, an increase in the number of channels is accompanied by a multifold increase in the number of parameters and in the computational complexity, lowering the cost effectiveness.

**Experiment 4:** Comparison of the performance of the network structures with different depths.

In DL, the network depth affects the performance and computational complexity of an algorithm. Here, network structures with various depths were designed based on model 2 (see Table 9) and compared in terms of the noise resistance and transferability. Table 10 summarizes the experimental results.

The experimental data show that model 2 with a five-layer structure was satisfactory in terms of the performance, number of parameters and computational complexity.

**Experiment 5:** Comparison of the transfer-learning ability for various levels of noise.

The transfer-learning ability of DL model 2, which had the best performance, was comparatively analyzed for various levels of noise. Table 11 summarizes the experimental results under different dataset combinations. For example, AB represents the test accuracy on sub-dataset B using a model trained by sub-dataset A.

The experimental data show that various noise intensity levels weakened the transfer-learning ability of model 2 to a certain extent (3–10%).

**Experiment 6:** Comparison of the noise resistance and transfer-learning ability of various algorithms.

To determine its noise resistance and transfer-learning ability, the proposed algorithm was compared with the available algorithms by using dataset A. Table 12 and Table 13 summarize the experimental results.

The experimental data show that the proposed algorithm exhibited the highest resistance to various noise intensity levels, 5–58% higher than that of the available algorithms.

The experimental data show that the proposed algorithm with the structure of model 2 exhibited the highest performance for various datasets, and its transfer-learning ability was 0.5–44% higher than that of the available algorithms.

### 4.3. Visualization of the Network Learning Process

It is impossible to illustrate how a CNN works because its working principle is similar to that of a black box. Therefore, the output characteristics of the network training process and the model testing process were subjected to a dimensionality reduction using the t-distributed stochastic neighbor embedding technique, and they were then visualized to facilitate the understanding of the operating process of the entire network.

Figure 10 shows the visualized network training process, demonstrating the feature distribution corresponding to the 0th to 500th iteration of the proposed algorithm on dataset A. As shown in Figure 10, as the number of times of the network training increases, the feature distance between the labels of different types increases, whereas the feature distance between the labels of the same type decreases, and they gradually form a cluster. This observation suggests that as the number of training iterations increases, the network’s ability to learn features and its classification accuracy improve.

Figure 11 shows the visualized network testing process, demonstrating the feature distribution corresponding to when the data pass through the conv_1 to conv_6 layers of the hybrid network. As shown in Figure 11, the features of all the samples are mixed and indistinguishable at the network initialization stage. As the number of network layers increases, the features gradually become distinguishable. In the conv_5 layer, the features of the same labels self-organize into one cluster.

## 5. Conclusions

This study investigates fault prediction in rotating machinery. Based on the ability of the wavelet packet transform to extract more frequency-domain information and based on the low computational load of 1D convolutions, a convolutional network-based lightweight deep learning fault prediction algorithm is proposed. This network can be regarded as having two layers. The first layer performs the wavelet packet transform (WPT), with an aim to extract finer information from a frequency-domain perspective, and the second layer is the designed relatively lightweight CNN (as shown in Figure 4). To validate the effectiveness of the proposed WPT-CNN network structure, the bearing dataset collected by the Case Western Reserve University (CWRU)’s Bearing Data Center were used to implement the experiments. The comparison results demonstrate that the proposed algorithm is superior to the available algorithms in noise resistance (5–58% higher) and transfer-learning ability (0.5–44% higher). In addition, the proposed algorithm also has the lowest computational complexity and memory space requirement (72.5% and 88.5% lower than those of the available algorithms, respectively). Therefore, the proposed algorithm can effectively improve the identification accuracy and is relatively lightweight.

Future research will consist in developing a new model for the deep convolution neural network optimization problem by improving the regularization term and optimization strategy with the results of this paper to predict the fault characteristics of the rotating machinery. Furthermore, to meet the actual working conditions of industrial production, future research will also focus on improving the generalization ability and the lightweight problem of the model.

## Figures and Tables

**Figure 1 sensors-19-02381-f001:**
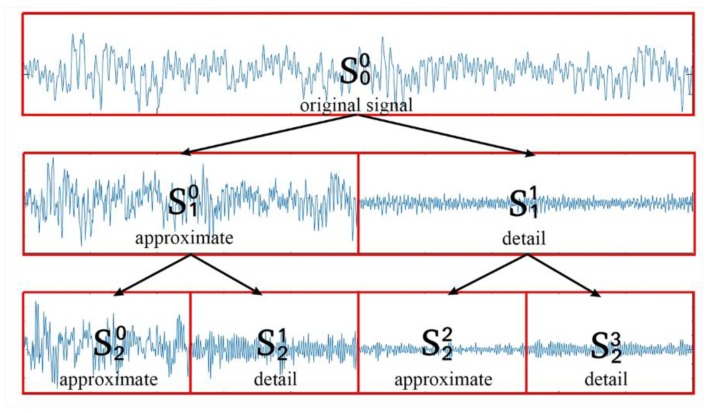
Tree structures of the wavelet packet transform.

**Figure 2 sensors-19-02381-f002:**
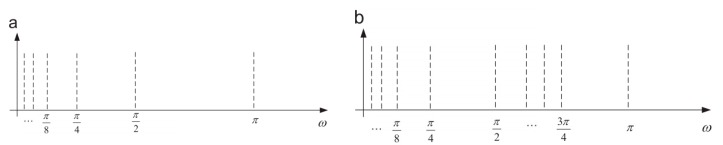
The advantages of WPT: (**a**) Dyadic WT time-frequency plane tiling; (**b**) Dyadic WPT time-frequency plane tiling.

**Figure 3 sensors-19-02381-f003:**
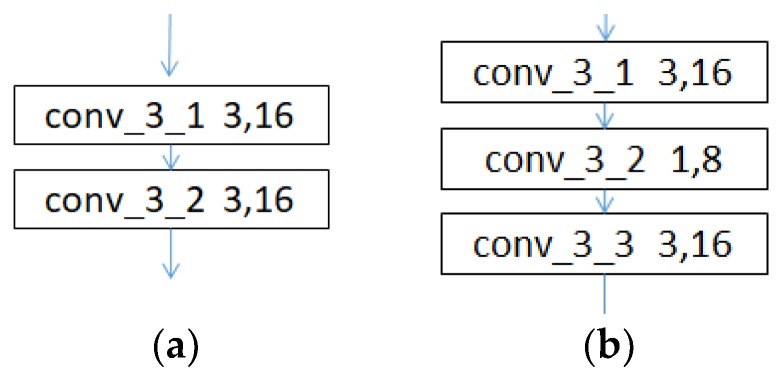
Comparison of different structures: (**a**) Dimensionality reduction without performing a 1 × 1 convolution; (**b**) Dimensionality reduction by performing a 1 × 1 convolution.

**Figure 4 sensors-19-02381-f004:**
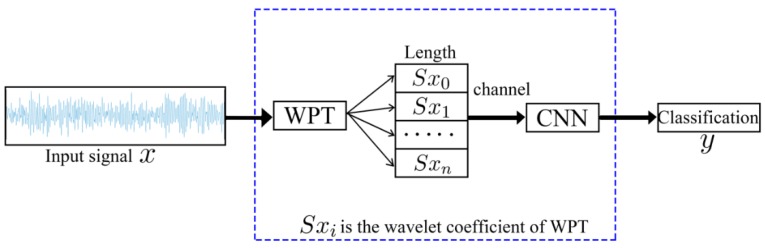
Lightweight CNN.

**Figure 5 sensors-19-02381-f005:**
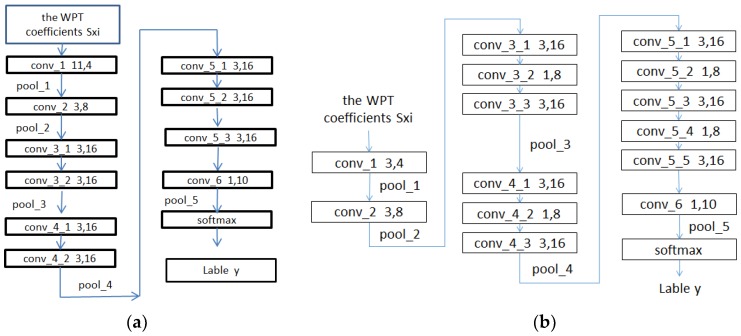
Structure of the CNN diagram: (**a**) The WPT-CNN network structures of model 1; (**b**) The WPT-CNN network structures of model 2.

**Figure 6 sensors-19-02381-f006:**
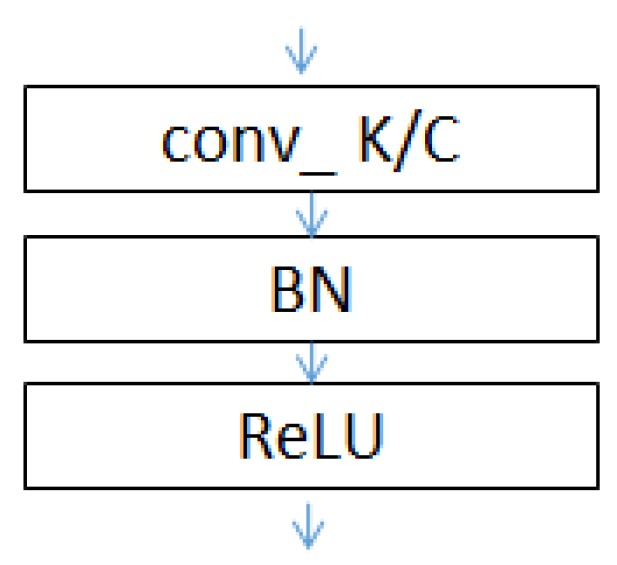
Structure of conv_i.

**Figure 7 sensors-19-02381-f007:**
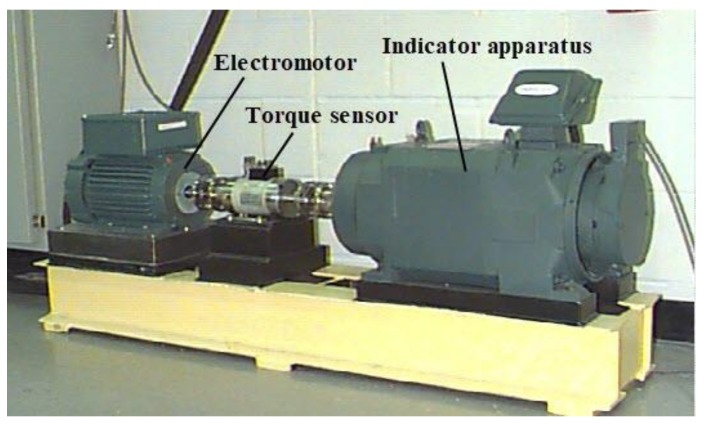
Simulation experimental platform for rolling bearing faults.

**Figure 8 sensors-19-02381-f008:**
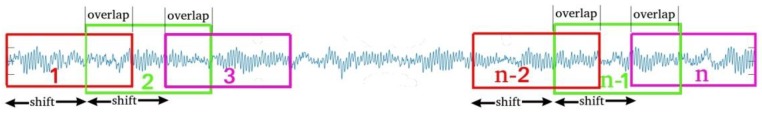
Schematic diagram of the data augmentation.

**Figure 9 sensors-19-02381-f009:**
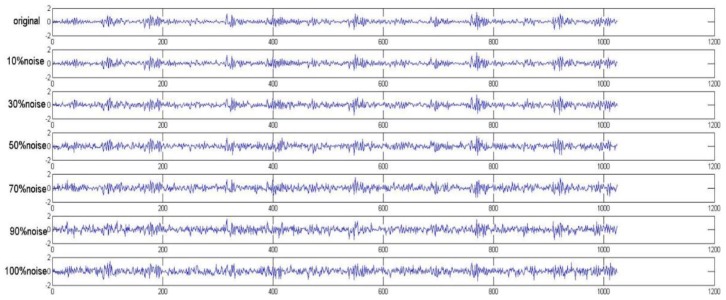
Original signal and signals with various percentages of added white Gaussian noise.

**Figure 10 sensors-19-02381-f010:**
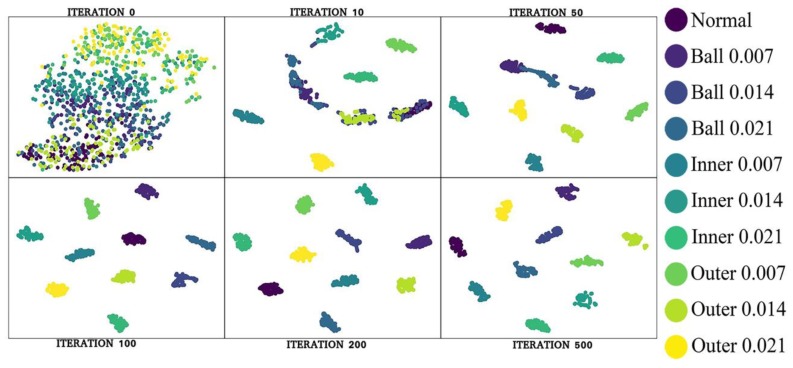
Feature distribution corresponding to the number of iterations.

**Figure 11 sensors-19-02381-f011:**
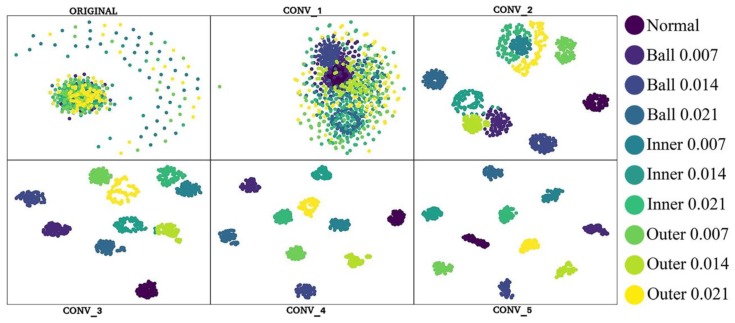
Feature distribution corresponding to number of layers.

**Table 1 sensors-19-02381-t001:** Comparison of computational resource requirements of the main available algorithms.

References	Memory Requirement	Computational Complexity
[28]	72.35 KB	1.40 × 10^6^
[23]	203.80 KB	6.81 × 10^7^
[24]	258.52 KB	4.69 × 10^6^
[25]	367.814 KB	2.00 × 10^8^
[27]	565.16 KB	8.08 × 10^7^
[15]	2696.54 KB	1.37 × 10^6^

**Table 2 sensors-19-02381-t002:** Number of channels and structures for various models.

	Model 1	Model 2	Model 3	Model 4	Model 5	Model 6	Model 7
	K/C	K/C	K/C	K/C	K/C	K/C	K/C
conv_1	11 × 1/4	11 × 1/4	11 × 1/4	11 × 1/4	11 × 1/8	11 × 1/16	11 × 1/32
conv_2	3 × 1/8	3 × 1/8	3 × 1/8	3 × 1/8	3 × 1/16	3 × 1/32	3 × 1/64
conv_3_1	3 × 1/16	3 × 1/16	3 × 1/16	3 × 1/16	3 × 1/32	3 × 1/64	3 × 1/128
conv_3_2	/	1 × 1/8	1 × 1/8	1 × 1/8	1 × 1/16	1 × 1/32	1 × 1/64
conv_3_3	3 × 1/16	3 × 1/16	3 × 1/16	3 × 1/16	3 × 1/32	3 × 1/64	3 × 1/128
conv_4_1	3 × 1/16	3 × 1/16	3 × 1/32	3 × 1/32	3 × 1/64	3 × 1/128	3 × 1/256
conv_4_2	/	1 × 1/8	1 × 1/16	1 × 1/16	1 × 1/32	1 × 1/64	1 × 1/128
conv_4_3	3 × 1/16	3 × 1/16	3 × 1/32	3 × 1/32	3 × 1/64	3 × 1/128	3 × 1/256
conv_5_1	3 × 1/16	3 × 1/16	3 × 1/32	3 × 1/64	3 × 1/128	3 × 1/256	3 × 1/512
conv_5_2	/	1 × 1/8	1 × 1/16	1 × 1/32	1 × 1/64	1 × 1/128	1 × 1/246
conv_5_3	3 × 1/16	3 × 1/16	3 × 1/32	3 × 1/64	3 × 1/128	3 × 1/256	3 × 1/512
conv_5_4	/	1 × 1/8	1 × 1/16	1 × 1/32	1 × 1/64	1 × 1/128	1 × 1/256
conv_5_5	3 × 1/16	3 × 1/16	3 × 1/32	3 × 1/64	3 × 1/128	3 × 1/256	3 × 1/512
conv_6	1 × 1/10	1 × 1/10	1 × 1/10	1 × 1/10	1 × 1/10	1 × 1/10	1 × 1/10
The number of parameters	23.77 KB	19.90 KB	56.93 KB	142.68 KB	557.07 KB	2201.10 KB	8750.16 KB
The computation of the floating numbers	1.77 × 10^5^	1.61 × 10^5^	2.66 × 10^5^	4.41 × 10^5^	1.57 × 10^6^	5.92 × 10^6^	2.29 × 10^7^
Note	Uncompressed 4 channels	Compressed 4 channels	Compressed 4 channels	Compressed 4 channels	Compressed 8 channels	Compressed 16 channels	Compressed 32 channels
	The maximum number of channels is 16	The maximum number of channels is 16	The maximum number of channels is 32	The maximum number of channels is 64	The maximum number of channels is 128	The maximum number of channels is 256	The maximum number of channels is 512

The WPT-CNN network structures of models 1 and 2 are shown in Figure 5a,b, respectively.

**Table 3 sensors-19-02381-t003:** Comparison of networks in terms of the parameters and computational load.

Method	Number of Parameters	Computation of Floating Number	Input Length Specification
WPT-CNN	19.90 KB	1.61 × 10^5^	64 × 1 × 16
Reference [28]	72.35 KB	1.40 × 10^6^	32 × 32 × 1
Reference [23]	203.80 KB	6.81 × 10^7^	400 × 1 × 1
Reference [24]	258.52 KB	4.69 × 10^6^	1024 × 1 × 1
Reference [27]	565.16 KB	8.08 × 10^7^	512 × 10 × 1
Reference [25]	367.814 KB	2.00 × 10^8^	128 × 128 × 1
Reference [15]	2696.54 KB	1.37 × 10^6^	1024 × 1 × 1
Resnet-50	80,849.75 KB	2.04 × 10^9^	1024 × 1 × 1
Resnet-18	14,325.75 KB	3.35 × 10^8^	1024 × 1 × 1
VGG-16	78,544.75 KB	9.24 × 10^8^	1024 × 1 × 1
1D-LeNet5	1349 KB	1.27 × 10^6^	1024 × 1 × 1

**Table 4 sensors-19-02381-t004:** The hardware and software environment.

Hardware Environment		Software Environment
CPU	Intel Core i7-8700k, 3.7 GHz, six-core twelve threads	Ubuntu 16.04 system TensorFlow 1.10 framework and Python programming language
GPU	NVIDIA 1080Ti 11G	
Memory	32 GB	
Storage	2 TB	

**Table 5 sensors-19-02381-t005:** Classification of the bearing fault datasets.

Datasets	Load (HP)	Training Samples	Test Samples	Fault Types	Flaw Size (Inches)	Models
A/B/C/D	0/1/2/3	800/800/800/800	100/100/100/100	normal	0	1
		800/800/800/800	100/100/100/100	ball	0.007	2
		800/800/800/800	100/100/100/100	ball	0.014	3
		800/800/800/800	100/100/100/100	ball	0.021	4
		800/800/800/800	100/100/100/100	inner_race	0.007	5
		800/800/800/800	100/100/100/100	inner_race	0.014	6
		800/800/800/800	100/100/100/100	inner_race	0.021	7
		800/800/800/800	100/100/100/100	outer_race	0.007	8
		800/800/800/800	100/100/100/100	outer_race	0.014	9
		800/800/800/800	100/100/100/100	outer_race	0.021	10

**Table 6 sensors-19-02381-t006:** Experimental results for the effects of the WPT level on the accuracy and noise resistance of the algorithm.

Validation Sets	Title 2	0%	10%	30%	50%	70%	90%	100%
Time domain	100	100	94.48	90.49	85.79	81.62	77.87	76.88
			±0.17	±0.57	±0.72	±1.15	±1.48	±0.88
WPT3	100	100	96.80	93.4	90.77	88.56	86.62	85.74
			±0.31	3±0.33	±0.33	±0.92	±0.71	±0.73
**WPT4**	**100**	**100**	**98.18**	**97.26**	**95.83**	**94.55**	**93.07**	**92.11**
			**±0.25**	**±0.37**	**±0.38**	**±0.47**	**±0.89**	**±0.63**
WPT5	100	99.4	96.69	95.78	94.30	92.66	90.40	90.16
			±0.26	±0.33	±0.49	±0.76	±0.69	±0.99
WPT6	100	97.2	96.02	94.03	91.49	89.01	86.43	85.83
			±0.34	±0.60	±0.84	±0.72	±1.10	±0.72

**Table 7 sensors-19-02381-t007:** Experimental results for the effects of the size of convolution kernels in the first layer on the noise resistance of the algorithm.

Convolution Kernels	Validation Sets	0%	10%	30%	50%	70%	90%	100%
3	100	100	98.18	97.26	95.83	94.55	93.07	92.11
			±0.25	±0.37	±0.38	±0.47	±0.89	±0.63
5	100	100	99.02	97.94	96.76	95.29	93.54	92.85
			±0.19	±0.32	±0.57	±0.39	±0.76	±0.42
7	100	99.9	99.73	99.11	98.44	96.98	95.31	95.03
			±0.18	±0.34	±0.41	±0.43	±0.54	±0.87
9	100	100	96.66	96.06	94.88	94.12	93.16	92.61
			±0.22	±0.39	±0.59	±0.51	±0.29	±0.92
11	100	100	99.33	99.02	98.63	97.71	96.57	96.35
			±0.12	±0.26	±0.24	±0.45	±0.43	±0.52
13	100	100	99.06	98.57	97.57	96.70	95.74	95.04
			±0.17	±0.34	±0.37	±0.52	±0.33	±0.45
15	100	100	99.50	99.08	98.38	97.49	96.35	96.24
			±0.17	±0.30	±0.21	±0.52	±0.57	±0.57
23	100	100	98.11	97.29	96.13	95.11	94.38	93.89
			±0.33	±0.27	±0.50	±0.53	±0.45	±0.59
31	100	99.9	99.03	98.28	97.30	96.47	95.52	94.70
			±0.14	±0.29	±0.41	±0.42	±0.55	±0.48

**Table 8 sensors-19-02381-t008:** Experimental results for the accuracy, noise resistance and transferability of various algorithms.

		Model 1	Model 2	Model 3	Model 4	Model 5	Model 6	Model 7
Noise resistance	0%	100	100	100	100	100	100	100
	10%	99.76	99.33	99.71	99.40	99.99	99.97	100.0
		±0.07	±0.12	±0.12	±0.18	±0.03	±0.05	±0.00
	30%	99.19	99.02	98.78	99.28	99.81	99.71	99.91
		±0.27	±0.26	±0.25	±0.27	±0.14	±0.14	±0.09
	50%	98.37	98.63	97.79	97.01	98.25	99.09	99.52
		±0.45	±0.24	±0.31	±0.60	±0.41	±0.34	±0.18
	70%	97.57	97.71	96.55	95.85	97.25	98.61	98.87
		±0.32	±0.45	±0.37	±0.37	±0.47	±0.29	±0.24
	90%	96.41	96.57	95.18	94.36	96.75	97.64	98.11
		±0.58	±0.43	±0.53	±0.84	±0.46	±0.40	±0.40
	100%	96.11	96.35	95.09	94.53	96.75	97.38	97.65
		±0.44	±0.52	±0.54	±0.42	±0.46	±0.20	±0.32
Transfer-learning ability	AB	98.8	100	99.9	100	98.4	99.2	99.4
	AC	99.1	100	100	99.9	99.6	99.8	97.3
	AD	99.3	99.9	99.5	99.9	98.7	99.7	99.2
Number of parameters		23.77 KB	19.90 KB	56.93 KB	142.68 KB	557.07 KB	2201.10 KB	8750.16 KB
The computation of floating numbers		1.77 × 10^5^	1.61 × 10^5^	2.66 × 10^5^	4.41 × 10^5^	1.57 × 10^6^	5.92 × 10^6^	2.29 × 10^7^

**Table 9 sensors-19-02381-t009:** Description of network structures with various depths.

	3 Layers	4 Layers	5 Layers	6 Layers
conv_1	11 × 1/4	11 × 1/4	11 × 1/4	11 × 1/4
conv_2	3 × 1/8	3 × 1/8	3 × 1/8	3 × 1/8
conv_3_1	3 × 1/16	3 × 1/16	3 × 1/16	3 × 1/16
conv_3_2	1 × 1/8	1 × 1/8	1 × 1/8	1 × 1/8
conv_3_3	3 × 1/16	3 × 1/16	3 × 1/16	3 × 1/16
conv_4_1		3 × 1/16	3 × 1/16	3 × 1/32
conv_4_2		1 × 1/8	1 × 1/8	1 × 1/16
conv_4_3		3 × 1/16	3 × 1/16	3 × 1/32
conv_5_1			3 × 1/16	3 × 1/16
conv_5_2			1 × 1/8	1 × 1/8
conv_5_3			3 × 1/16	3 × 1/16
conv_5_4			1 × 1/8	1 × 1/8
conv_5_5			3 × 1/16	3 × 1/16
conv_6_1				3 × 1/16
conv_6_2				1 × 1/8
conv_6_3				3 × 1/16
conv_6_4				1 × 1/8
conv_6_5				3 × 1/16
conv_6	1 × 1/10	1 × 1/10	1 × 1/10	1 × 1/10

**Table 10 sensors-19-02381-t010:** Comparison of the noise resistance and transferability of the network structures with various depths.

Layers		3	4	5	6
Validation sets		100	100	100	100
Normal		100	100	100	100
Transfer-learning ability	AB	90.9	99.70	100	99.6
	AC	93.5	99.40	100	99.8
	AD	91.5	98.90	99.8	99.6
Noise resistance	10%	99.11 ± 0.40	99.72 ± 0.10	99.33 ± 0.12	98.95 ± 0.17
	30%	97.05 ± 0.40	99.32 ± 0.26	99.02 ± 0.26	98.32 ± 0.32
	50%	94.33 ± 0.60	98.69 ± 0.25	98.63 ± 0.24	97.29 ± 0.58
	70%	91.87 ± 0.74	97.49 ± 0.36	97.71 ± 0.45	95.69 ± 0.43
	90%	89.35 ± 0.95	96.61 ± 0.67	96.57 ± 0.43	94.42 ± 0.51
	100%	87.94 ± 0.96	95.88 ± 0.59	96.35 ± 0.52	93.97 ± 0.55
Number of parameters		7.49 KB	12.65 KB	19.90 KB	27.15 KB
The computation of floating number		1.26 × 10^5^	1.47 × 10^5^	1.61 × 10^5^	1.75 × 10^5^

**Table 11 sensors-19-02381-t011:** Comparison of the transfer-learning ability of the proposed algorithm in terms of noise resistance.

	0%	10%	30%	50%	70%	90%	100%
AB	100	99.54 ± 0.25	98.82 ± 0.24	97.71 ± 0.47	96.96 ± 0.50	96.12 ± 0.57	95.53 ± 0.51
AC	100	99.42 ± 0.14	98.78 ± 0.25	98.21 ± 0.32	97.38 ± 0.45	96.64 ± 0.44	95.94 ± 0.38
AD	99.8	97.21 ± 0.21	96.35 ± 0.43	95.63 ± 0.46	94.64 ± 0.63	93.16 ± 0.47	93.28 ± 0.88
BA	98.9	97.07 ± 0.33	96.16 ± 0.43	94.42 ± 0.48	93.13 ± 0.51	91.79 ± 1.00	90.34 ± 0.73
BC	100	99.47 ± 0.15	99.44 ± 0.15	99.19 ± 0.18	98.54 ± 0.29	98.11 ± 0.42	97.65 ± 0.42
BD	99.9	98.15 ± 0.22	97.78 ± 0.36	97.23 ± 0.20	96.66 ± 0.30	95.64 ± 0.32	95.52 ± 0.61
CA	93.4	92.81 ± 0.31	91.40 ± 0.38	89.99 ± 0.62	88.59 ± 0.82	86.93 ± 0.80	86.31 ± 0.53
CB	98.7	97.27 ± 0.17	96.87 ± 0.34	96.54 ± 0.40	95.86 ± 0.39	95.44 ± 0.62	94.88 ± 0.27
CD	100	99.87 ± 0.13	99.50 ± 0.13	99.01 ± 0.27	98.48 ± 0.36	97.58 ± 0.52	97.23 ± 0.42
DA	95.3	91.84 ± 0.32	90.45 ± 0.54	89.18 ± 0.56	88.41 ± 0.58	86.90 ± 0.60	86.33 ± 0.76
DB	98.4	97.98 ± 0.14	97.59 ± 0.33	97.31 ± 0.22	96.73 ± 0.48	95.99 ± 0.39	95.90 ± 0.51
DC	100	99.72 ± 0.04	99.66 ± 0.15	98.97 ± 0.17	99.05 ± 0.14	98.19 ± 0.32	97.88 ± 0.29

**Table 12 sensors-19-02381-t012:** Resistance of various algorithms to various percentages of added white Gaussian noise.

	Validation Sets	0%	10%	30%	50%	70%	90%	100%
WPT-CNN	100	**100**	99.33	**99.02**	**98.63**	**97.71**	**96.57**	**96.35**
			±0.12	**±0.26**	**±0.24**	**±0.45**	**±0.43**	**±0.52**
Reference [28]	100	**100**	**99.85**	98.33	95.60	91.95	88.02	85.94
			**±0.08**	±0.27	±0.44	±0.52	±0.55	±1.13
Reference [24]	100	99.70	99.53	98.24	94.51	90.72	86.69	85.35
			±0.18	±0.26	±0.61	±0.51	±0.54	±0.45
Reference [23]	100	100	80.13	71.61	64.50	59.88	57.01	55.83
			±0.19	±0.27	±0.44	±0.37	±0.40	±0.37
Reference [26]	100	100	88.10	66.28	55.24	48.38	43.89	42.02
			±0.87	±0.58	±1.20	±0.90	±1.03	±0.98
Reference [27]	100	99.90	97.46	85.14	70.59	58.57	50.96	48.38
			±0.52	±0.80	±0.75	±1.03	±0.47	±0.73
Reference [19]	100	99.80	99.15	98.01	96.36	94.03	91.79	90.74
			±0.20	±0.50	±0.31	±0.69	±0.86	±0.64
Reference [15]	99.1	97.50	95.54	93.88	92.86	90.81	88.90	87.49
			±0.25	±0.50	±0.38	±0.66	±1.03	±0.92
Reference [11]	93.6	88.30	88.08	85.38	76.66	64.03	46.81	38.47
			±0.33	0.77	±0.88	±0.75	±1.37	±0.95

**Table 13 sensors-19-02381-t013:** Comparison of the transfer-learning ability of various algorithms.

	AB	AC	AD	BA	BC	BD	CA	CB	CD	DA	DB	DC	AVG
WPT-CNN	**100**	**100**	**99.8**	98.9	**100**	**99.9**	93.4	98.7	**100**	95.3	**98.4**	**100**	**98.70**
Reference [28]	98.4	99.9	98.2	**99.8**	100	98.9	96.0	98.3	99.3	90.4	93.8	100	97.75
Reference [24]	99.5	99.9	96.9	99.5	100	98.2	96.8	98.4	99.0	**97.1**	97.1	99.1	98.46
Reference [23]	99.9	98.9	89.8	99.9	99.9	97.9	**98.6**	**99.4**	**99.8**	81.3	85.0	94.2	95.40
Reference [26]	89.3	84.5	73.0	83.6	98.0	91.5	85.1	96.9	95.9	78.0	88.0	95.3	88.26
Reference [27]	98.7	98.3	81.7	98.7	99.9	98.5	93.2	97.7	97.4	89.4	92.0	94.9	95.03
Reference [19]	71.1	74.8	72.6	87.6	87.4	76.6	77.5	88.0	79.5	77.7	78.9	86.7	79.87
Reference [15]	33.7	44.6	43.6	43.3	42.0	51.5	48.4	47.7	48.3	42.8	46.3	41.8	44.50
Reference [11]	40.4	40.3	39.3	40.5	55.3	59.8	41.2	56.3	57.1	36.9	56.1	52.6	47.98

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
