# Peer review of "A Lighted Deep Convolutional Neural Network Based Fault Diagnosis of Rotating Machinery"

_sensors, 2019, doi:10.3390/s19102381_

Reviewer 1 Report

In this paper, a fault diagnosis method for rotating machinery was presented. The key idea is to use wavelet packet transfer to get subband coefficients, then use these coefficients as the input of deep convolutional network (DCN). To address the computational complexity, a light weight 1D convolutional neural network (CNN) was proposed. In general, this is an interesting research and the paper is easy to follow. However, the paper should be improved in the following aspects:

How the proposed algorithm was implemented? Did authors verify the implementation? In reference [27], the proposed DRN+DWWC method was implemented using TensorFlow, which is Google’s open source software library for machine learning and verified by two datasets.

The dataset used in the paper can't be accessed now? How did authors get the dataset?

In Table 2, 6 Models were listed and the Figure 5 shows how to implement the Model 1 and 2 into the proposed WPT-CNN. Based on the data shown in Table 2, It seems like Model 2 is better than Model 1 (line 210-211). Which Model was used in the performance comparison with other researcher's result?

As it was mentioned by authors, the initial value has certain impact to the training and system performance of the network (line 254-255). Why did authors select the Xavier initialization? Any justification?

In summary, without the information of algorithm implementation and verification, the reviewer is hard to be convinced by comparison results. Therefore, this paper can't be considered for acceptance at the current stage.

Author Response

Response to Reviewer 1 Comments

Point 1: How the proposed algorithm was implemented? Did authors verify the implementation? In reference [27], the proposed DRN+DWWC method was implemented using TensorFlow, which is Google’s open source software library for machine learning and verified by two datasets.

Response 1: As the reviewer said, this paper also uses TensorFlow to implement the algorithm. The bearing dataset collected by the Case Western Reserve University (CWRU)'s Bearing Data Center were used to validate the proposed algorithm.

[37] Data sets. Available online: http://csegroups.case.edu/bearingdatacenter/pages/download-data-file. (accessed on 11 May 2013).

Point 2: The dataset used in the paper can't be accessed now? How did authors get the dataset?

Response 2: Perhaps the previous website is now inaccessible. We have updated the source address of the dataset (http://csegroups.case.edu/bearingdatacenter/pages/download-data-file), which can be downloaded from the new link.

Point 3: In Table 2, 6 Models were listed and the Figure 5 shows how to implement the Model 1 and 2 into the proposed WPT-CNN. Based on the data shown in Table 2, It seems like Model 2 is better than Model 1 (line 210-211). Which Model was used in the performance comparison with other researcher's result?

Response 3: In this paper, seven models were designed (as shown in Table 2, the previous version omitted the Model 7 which has been added in revised manuscript), which represent the different numbers of channels and structures in the CNN part. The results show that model 2 is optimal. The relative description has been supplemented in Section 3.1.

Point 4: As it was mentioned by authors, the initial value has certain impact to the training and system performance of the network (line 254-255). Why did authors select the Xavier initialization? Any justification?

Response 4: Weight initialization of deep learning model has an effect on training. If the weight initialization of the deep learning model is too small, when signals are transmitted between layers, the signal will gradually decay and it is difficult to produce effect. If the weight initialization is too large, the signal will gradually amplify and cause divergence and invalid. The function of Xavier initializer is not only to make the weight more suitable when initializing the deep learning network, but also to accelerate the convergence of the network. These advantages have been verified by Glorot [38]. In order to facilitate the reader to understand, reference [38] has been supplemented.

[38] Glorot, X,; Bengio, Y. Understanding the difficulty of training deep feedforward neural networks. 13th International conference on artificial intelligence and statistics, Sardinia, Italy, 13-15 May 2010.

Point 5: In summary, without the information of algorithm implementation and verification, the reviewer is hard to be convinced by comparison results. Therefore, this paper can't be considered for acceptance at the current stage.

Response 5: The algorithm proposed in this paper has been implemented using TensorFlow, and the relevant verification and comparison with other algorithms have also been performed such as comparison of networks in terms of parameters and computational load (as shown in Table 3), comparison of the noise resistance and transfer-learning ability of various algorithms (as shown in Table 11) and comparison of the transfer-learning ability of various algorithms (as shown in Table 12).

Reviewer 2 Report

The paper presents a method for fault diagnosis of rotating machinery. The method include CNN and wavelet transform.
The content of the paper is sufficient for publication. The paper also presented the comparison between the proposed method and previous published articles.
However, there are few comments to improve the paper prior to publication:
1. In Section 1, the paper provided the comparison of computational resource requirements from few published articles as presented in Table 1. I am curious to know how the Authors calculated the computational complexity?
2. In Section 2, the WPT method basically decomposes a raw vibration signal into two different levels namely approximate and detail. Please indicate these component in Figure 1.
3. Please enhance the resolution of Figure 4. The input signal can be revised to better resolution and increase the font size of Figure 4.
4. It would be better of the WPT-CNN structure described in Line 213 to 253 is also presented in flowchart. This will make reader easily understand the flow process.
5. Name the features presented in Figure 10. Different color is refer to particular feature.
6. The Conclusion need to be added with more sentences. Probably the future work such as validate the proposed method using the real fault bearing data can also be added.

Author Response

Response to Reviewer 2 Comments

Point 1: In Section 1, the paper provided the comparison of computational resource requirements from few published articles as presented in Table 1. I am curious to know how the Authors calculated the computational complexity?

Response 1: There are two main criteria to measure the complexity of the algorithm:

(1)  Number of parameters

The parameter length of each model is 32 bit floating point number, which occupies the storage space of 4Byte. The calculation methods of the number of parameters in the convolutional layer and the fully connected layer are shown in Eqs. (9) and (10), respectively. The number of parameters is the sum of the number of parameters of each network layer, and the number of parameters of the model represents the size of storage space occupied by all parameters.

(2)  The computation of floating numbers

The computation of floating numbers is expressed by the number of floating-point operations that each network layer propagates forward to process one data. The realization of convolution or full connection is actually one multiplication and one addition. The calculation methods are shown in Eqs. (11) and (12). The computation of floating numbers of the model is the sum of all network layers.

Point 2: In Section 2, the WPT method basically decomposes a raw vibration signal into two different levels namely approximate and detail. Please indicate these component in Figure 1.

Response 2: The reviewer's comments are correct, the information description about Approximate and Detail of signals have been supplemented in Figure 1.

Point 3: Please enhance the resolution of Figure 4. The input signal can be revised to better resolution and increase the font size of Figure 4.

Response 3: According to reviewer's comment, Figure 4 has been revised to better resolution and increase the font size.

Point 4: It would be better of the WPT-CNN structure described in Line 213 to 253 is also presented in flowchart. This will make reader easily understand the flow process.

Response 4: Lines 213 to 253 describe the technical details of the WPT-CNN network structure, the purpose is to focus on how to select parameters in the process of algorithm implementation. For example, 2. The lengths of the conv_1 to conv_5 convolutional layers of the network are different, and their values are set to 11, 3, 3, 3 and 3, respectively. These technical details are difficult to express in the form of flow charts.

Point 5: Name the features presented in Figure 10. Different color is refer to particular feature.

Response 5: The reviewer's comments are correct, different color is refer to particular feature. Figures 10 and 11 have been revised, and the fault categories corresponding to each color are supplemented.

Point 6: The Conclusion need to be added with more sentences. Probably the future work such as validate the proposed method using the real fault bearing data can also be added.

Response 6: According to reviewer's comment, the conclusion has been rewritten, and the future work has also been added.

Conclusions:

This study investigates fault prediction in rotating machinery. Based on the ability of the wavelet packet transform to extract more frequency-domain information and the low computational load of 1D convolutions, a convolutional network-based lightweight deep learning fault prediction algorithm is proposed. This network can be regarded as two layers. The first layer performs the wavelet packet transform (WPT), with an aim to extract finer information from a frequency-domain perspective, and the second layer is the designed relatively lightweight CNN (as shown in Figure 4). To validate the effectiveness of the proposed WPT-CNN network structure, the bearing dataset collected by the Case Western Reserve University (CWRU)'s Bearing Data Center were used to implement experiments. The comparison results demonstrate that the proposed algorithm is superior to the available algorithms in noise resistance (5-58% higher) and transfer-learning ability (0.5-44% higher). In addition, the proposed algorithm also has the lowest computational complexity and memory space requirement (72.5% and 88.5% lower than those of the available algorithms, respectively). Therefore, the proposed algorithm can effectively improve the identification accuracy and is relatively lightweight.

Future research will consist of developing a new model for the deep convolution neural network optimization problem by the improvement of regularization term and optimization strategy with the results of this paper to predict the fault characteristics of the rotating machinery. Also, to meet the actual working conditions of industrial production, future research will also focus on improving the generalization ability and the lightweight problem of the model.

Reviewer 3 Report

The manuscript relates to the problem of fault diagnosis of rotating machinery. Considering the fact that rotating machine parts constitute very significant and numerous group of machine elements the subject of the manuscript is a matter of great importance (for example bearings). Authors propose the algorithn that consists of two main steps - the former is a pyramid wavelet packet decomposition pf the vibration signal. Then latter is to apply a a deep convolutional network (DCN). Generally I think that the manuscript is very interesting. I have only a few minor comments:

I think that the method proposed by authors can be applied to fault diagnosis of bearings. In the introduction authors do not mention about methods based on measurements of anti-torque of bearings, which was for example described in work: 

Adamczak S., Stępień K., Wrzochal M. Comparative Study of Measurement Systems Used to Evaluate Vibrations of Rolling Bearings, Procedia Engineering, Volume 192, 2017, Pages 971-975

Authors propose to apply wavelet pocket decomposition to investigate rotary elements. . Some works refer to similar problem, for example in work:

StÄ™pieÅ„ K., MakieÅ‚a W.: An analysis of deviations of cylindrical surfaces with the use of wavelet transform , Metrology and Measurement Systems, Vol. XX (2013), No. 1, pp. 139−150. I think it would be interesting if authors could refer to this work

As it is know wavelet transform is applied with the use of so-called mother wavelet. Authors apply Daubechieu  -1 (which is in fact Haar wavelet). Have authors tried to apply some other types of mother wavelets?

Author Response

Point 1: I think that the method proposed by authors can be applied to fault diagnosis of bearings. In the introduction authors do not mention about methods based on measurements of anti-torque of bearings, which was for example described in work:

Adamczak S., Stępień K., Wrzochal M. Comparative Study of Measurement Systems Used to Evaluate Vibrations of Rolling Bearings, Procedia Engineering, Volume 192, 2017, Pages 971-975

Authors propose to apply wavelet pocket decomposition to investigate rotary elements. Some works refer to similar problem, for example in work:

Stępień K., Makieła W.: An analysis of deviations of cylindrical surfaces with the use of wavelet transform , Metrology and Measurement Systems, Vol. XX (2013), No. 1, pp. 139−150. I think it would be interesting if authors could refer to this work

Response 1: According to reviewer's comment, the references mentioned above have been added in revised manuscript.

Point 2: As it is know wavelet transform is applied with the use of so-called mother wavelet. Authors apply Daubechieu  -1 (which is in fact Haar wavelet). Have authors tried to apply some other types of mother wavelets?

Response 2: We have tried many kinds of wavelets, from db1 to db10. The experimental results show that the type of mother wavelet has little effect on the accuracy, noise resistance and migration of the network structure proposed in this paper. We chose to use db1 to reduce the computational complexity of the network because db1 does not increase the length of subband coefficients.

Round  2

Reviewer 1 Report

In the reply to reviewer, authors addressed my comments and concerns. However, the revision has not been fully revised based on the reply. Clearly mention the algorithm implementation and verification in the manuscript is required. The newly provided dataset link works. Which dataset was used for comparison? The dataset access date should be updated.

Author Response

Point 1: In the reply to reviewer, authors addressed my comments and concerns. However, the revision has not been fully revised based on the reply. Clearly mention the algorithm implementation and verification in the manuscript is required. The newly provided dataset link works. Which dataset was used for comparison? The dataset access date should be updated.

Response 1: The hardware and software environment used to implement the algorithm have been supplemented in Section 4 (as shown in Table 4).

Table 4. The hardware and software environment.

Hardware   environment

Software   environment

CPU

Intel   Core i7-8700k, 3.7GHz, six-core twelve threads

Ubuntu   16.04 system

TensorFlow   1.10 framework and Python programming language

GPU

NVIDIA   1080Ti 11G

Memory

32 GB

Storage

2 TB

A, B, C and D represents four types of faults corresponding to the data collected under 0, 1, 2 and 3 loads, respectively. For example, A subdataset denotes the data collected under load 0. As shown in Table 11 and Table 13, all datasets have been used for comparison of transfer-learning ability, for example, AB represents the test accuracy on subdataset B using a model trained by subdataset A. To facilitate the reader understanding, some additional explanations for datasets in Section 4.1.1, experiment 5 and experiment 6.

Also, the dataset access date has been updated.

Round  3

Reviewer 1 Report

Authors have clearly addressed my comments. No more comments.